# Hyperosmotic Stress Induces Phosphorylation of CERT and Enhances Its Tethering throughout the Endoplasmic Reticulum

**DOI:** 10.3390/ijms23074025

**Published:** 2022-04-05

**Authors:** Kentaro Shimasaki, Keigo Kumagai, Shota Sakai, Toshiyuki Yamaji, Kentaro Hanada

**Affiliations:** 1Department of Biochemistry and Cell Biology, National Institute of Infectious Diseases, 1-23-1 Toyama, Shinjuku-ku, Tokyo 162-8640, Japan; shimak@niid.go.jp (K.S.); sakais@niid.go.jp (S.S.); tyamaji@niid.go.jp (T.Y.); 2Department of Quality Assurance, Radiation Safety and Information System, National Institute of Infectious Diseases, 1-23-1 Toyama, Shinjuku-ku, Tokyo 162-8640, Japan

**Keywords:** lipid transfer protein, regulation, sphingomyelin, VAP, membrane contact sites, very-long-chain

## Abstract

The ceramide transport protein (CERT) delivers ceramide from the endoplasmic reticulum (ER) to the Golgi apparatus, where ceramide is converted to sphingomyelin (SM). The function of CERT is regulated in two distinct phosphorylation-dependent events: multiple phosphorylations in a serine-repeat motif (SRM) and phosphorylation of serine 315 residue (S315). Pharmacological inhibition of SM biosynthesis results in an increase in SRM-dephosphorylated CERT, which serves as an activated form, and an enhanced phosphorylation of S315, which augments the binding of CERT to ER-resident VAMP-associated protein (VAP), inducing the full activation of CERT to operate at the ER–Golgi membrane contact sites (MCSs). However, it remains unclear whether the two phosphorylation-dependent regulatory events always occur coordinately. Here, we describe that hyperosmotic stress induces S315 phosphorylation without affecting the SRM-phosphorylation state. Under hyperosmotic conditions, the binding of CERT with VAP-A is enhanced in an S315 phosphorylation-dependent manner, and this increased binding occurs throughout the ER rather than restrictedly at the ER–Golgi MCSs. Moreover, we found that de novo synthesis of SM with very-long acyl chains preferentially increases via a CERT-independent mechanism under hyperosmotic-stressed cells, providing an insight into a CERT-independent ceramide transport pathway for de novo synthesis of SM.

## 1. Introduction

Intracellular trafficking of lipids is an essential event for the metabolism of lipids and membrane biogenesis in cells. In eukaryotic cells, the endoplasmic reticulum (ER) is the center of de novo synthesis of various lipid types, and lipid transport mediated by lipid transfer proteins (LTPs) at organelle membrane contact sites (MCSs) is the predominant mechanism to transport lipid from the ER to other organelles [1,2]. However, how the function of LTPs and the formation of MCSs are regulated in cells remains poorly understood.

In the synthesis of sphingomyelin (SM) in the Golgi apparatus, the ceramide transport protein (CERT), a typical LTP, delivers the precursor ceramide synthesized in the ER to the Golgi apparatus, where ceramide is converted to SM [3]. CERT contains a steroidogenic acute regulatory protein-related lipid transfer (START) domain, which catalyzes the inter-membrane transfer of ceramide as its lipid-transfer domain. In addition, to execute rapid and accurate ceramide transport at the ER–Golgi contact sites, CERT possesses the pleckstrin homology (PH) domain that preferentially recognizes phosphatidylinositol 4-monophosphate [PtdIns(4)P], which is the predominant phosphoinositide in the Golgi membranes, and two phenylalanines in an acidic tract (FFAT) motif that associates with VAMP-associated protein (VAP), an ER-resident protein [4]. Of note, de novo synthesis of SM marginally occurs even in the absence of CERT [3,5,6], suggesting the existence of a CERT-independent pathway for ER-to-Golgi transport of ceramide, which remains poorly characterized.

The function of CERT is regulated in at least two distinct phosphorylation-dependent events. One of the phosphorylation sites is a serine-repeat motif (SRM), which resides downstream of the PH domain. Once the first serine (S132) in the SRM is phosphorylated by protein kinase D [7], the subsequent serine/threonine residues are sequentially phosphorylated by casein kinase 1ɤ [8]. When the SRM receives multiple phosphorylations, the activities of the PH domain, FFAT motif, and START domain are simultaneously repressed, and the function of CERT is consequently inactivated [9,10]. The S132A mutant, which mimics the dephosphorylated SRM form of CERT, exhibits a constitutively active feature [7,9]. Another phosphorylation site in CERT is serine 315 (S315), which resides upstream of the FFAT motif. The phosphorylation of S315 results in an increase in the FFAT motif-dependent affinity of CERT for VAP, enhancing the ER-to-Golgi ceramide trafficking function of CERT [10], although it remains unclear whether CERT is capable of binding to VAP anywhere in the ER or at the ER–Golgi MCSs. When de novo synthesis of SM is pharmacologically blocked, the SRM of CERT is dephosphorylated while S315 is phosphorylated [10], providing evidence that the two events of the SRM dephosphorylation and S315 phosphorylation are coordinated to fully activate CERT when the cells require SM. However, it remains unknown whether these two events are always coordinated.

In the present study, we found that hyperosmotic stress induces the phosphorylation of CERT S315 residue while the phosphorylation state of SRM remains multiply phosphorylated, which indicates that the two phosphorylation events can occur independently. Under hyperosmotic conditions, the interaction of CERT with VAP-A is enhanced in a CERT S315 phosphorylation-dependent manner, and this increased binding occurs throughout the ER rather than at the ER –Golgi MCSs. In addition, we unexpectedly found that de novo synthesis of SM with very-long acyl chains increases in a CERT-independent manner in hyperosmotic stressed cells, providing a feature of a CERT-independent ceramide transport pathway for de novo synthesis of SM.

## 2. Results

### 2.1. Phosphorylation State of S315 but Not SRM in CERT Is Affected by Hyperosmotic Stress

In this study, to employ retroviral transfection, we used the HeLa mCAT#8 cell line, which stably expresses the mouse ecotropic retroviral receptor mCAT-1. We employed a HeLa CERT knockout (KO) cell line to prevent any interference with the endogenous wild-type (WT) CERT. In addition, when various CERT variants were expressed, CERT KO/shCERT cells that stably express a short hairpin RNA matching to human CERT were used as the parent HeLa KO cells to prevent overexpression artifacts unless otherwise noted. Using the cell system, we could control the protein expression levels of CERT variants in the CERT KO background near to the endogenous CERT level in wild-type HeLa cells (Figure 1A). To examine whether the phosphorylation states of CERT are altered in response to hyperosmotic stress, we used 10% serum-containing culture media supplemented with various concentrations of sorbitol as hyperosmotic media unless otherwise noted. Upon incubation of cells in a 600 mM sorbitol-supplemented medium for 60 min, the phosphorylation level of S315 in HA-CERT wild-type (WT) was markedly elevated (Figure 1B). This elevation was not due to any non-specific reactivity of the antibody used because no signal was detected in cells expressing HA-CERT S315A, which is incapable of being phosphorylated at the S315 position (Figure 1B). In addition, Western blotting for the HA-tag showed that hyperosmotic stress did not affect the protein expression level of HA-CERT, ruling out the possibility that the elevated signal of S315 is due to increased expression of CERT (Figure 1B). Notably, the doublet pattern of CERT did not discernibly alter (Figure 1B). In the doublets, the upper band represents multiply phosphorylated forms in the SRM, while the lower band represents hypo-phosphorylated and/or de-phosphorylated form(s) [9,10]. The minor possibility that the upper band of the CERT doublet in cells exposed to the hyperosmotic stress did not represent the SRM-hyperphosphorylated form was eliminated because protein phosphatase treatment of cell lysate fractions shifted the upper band to the lower band independent of whether there was or was not an osmotic stimulus (Figure 1C). Thus, the characterization of the phosphorylation states of CERT within cells exposed to hyperosmotic stress let us notice that the events of SRM dephosphorylation and the S315 phosphorylation might not necessarily be coordinated. To detect the S315 phosphorylated form by Western blotting analysis with anti-CERT S315p polyclonal antibodies, we had to concentrate CERT from the cell lysate fraction by immunoprecipitation (Figure 1D). HA-CERT was immunoprecipitated with an anti-HA-epitope monoclonal antibody but not with commercially available anti-CERT polyclonal antibodies. Thus, for this technical reason, we could not verify that the hyperosmotic stress affected the level of phosphorylation of S315 of the endogenous CERT.

A previous study using CERT-overexpressing HeLa cells showed that when cells were treated with ISP-1/myriocin, a potent inhibitor of serine palmitoyltransferase (the enzyme catalyzing the first step in de novo synthesis of all sphingolipid types), S315 of HA-CERT WT was shifted to be phosphorylated [10]. In agreement with this previous study, the S315 phosphorylation was discernibly enhanced upon ISP-1-treatment with HeLa cells expressing HA-CERT at a near endogenous level, but the elevated level was far less than that seen in response to the hyperosmotic treatment (Figure 1E). In contrast with the case of the hyperosmotic treatment, the SRM dephosphorylation form (the lower band of the doublet) increased in response to ISP-1 treatment, in line with previous studies [9,10]. These results indicate that responses in the phosphorylation states of the SRM and S315 of CERT vary depending on stimulus types.

### 2.2. Severe Hyperosmotic Stress, but Not Mild Hyperosmotic nor Hypoosmotic Stress, Induces Reversible CERT S315 Phosphorylation

We next examined whether the elevation of CERT S315 phosphorylation depended on the strength and duration of hyperosmotic stress. When cells were treated with various concentrations (100–800 mM) of sorbitol, and for different time periods, the phosphorylation states of the SRM were not altered (Figure 1F). In contrast, the phosphorylation states of S315 clearly increased in a sorbitol concentration and exposure period-dependent manners (Figure 1F): At 800 mM sorbitol, the level of the S315 phosphorylation gradually increased and reached a plateau at 60 min. At 400 mM sorbitol, the level of the S315 phosphorylation peaked at 30 min—but the degree of the peak was weaker than that of 800 mM sorbitol—and then reduced to the untreated control level. Significant changes were not observed at 100 mM sorbitol. Importantly, when NaCl was used as an osmolyte in place of sorbitol, the phosphorylation of CERT S315 was also elevated, ruling out the possibility that the elevation of the S315 phosphorylation was a sorbitol-specific event (Figure 1G).

The osmolarity of the medium supplemented with 800 mM sorbitol is about 1100 mOsm, which should burden cells with severe hyperosmotic stress. This invoked the concern that the enhanced S315 phosphorylation was simply an irreversible event occurring in dying cells. However, the S315 phosphorylation elevation was found to be reversible (Figure 2A): When cells were cultured in isosmotic media after exposure to 800 mM sorbitol for 60 min, the S315 phosphorylation level of the cells gradually returned to the near-basal level (Figure 2A). These results demonstrate that the elevation of the CERT S315 phosphorylation in response to the severe hyperosmotic stress is not a response occurring only in dying cells.

We also tested whether the phosphorylation states of CERT were responsive to hypoosmotic stress. When cells were exposed to a 1.5-fold diluted culture medium for various time periods, the states of multiply phosphorylated SRM and the S315 phosphorylation in CERT were unchanged (Figure 2B). A previous study showed that hyperosmotic stress (550 mOsm for 48 h) induced the upregulation of SM synthesis in MDCK cells [11]. Therefore, we challenged the cells’ prolonged exposure under relatively mild hyperosmotic conditions (500 mOsm for 24 h). However, both the states of multiply phosphorylated SRM and the S315 phosphorylation did not change (Figure 2B).

### 2.3. The FFAT Motif-Dependent Interaction of CERT with VAP-A Is Enhanced under Hyperosmotic Conditions

We next examined impacts of hyperosmotic stress on the CERT–VAP interaction by a co-precipitation assay. When cells expressing HA-CERT WT at a nearly endogenous level were used, we failed to detect any discernible co-precipitation signal. Thus, we employed new cell lines in which CERT variants with an *N*-terminal mVenus-tag were ectopically expressed within the parental HeLa KO cell line (Figure 3A). In these cells, two bands with apparent molecular masses of ~100 kDa were detected using anti-CERT and anti-GFP antibodies, and these two bands were also detected with an anti-*C* terminus CERT antiserum (Figure 3A), suggesting that they are the products translated from two possible initiation codons in the open-reading frame of the *N*-terminally tagged-mVenus. The cells were incubated in the isosmotic or hyperosmotic medium for 1 h and treated with the membrane-permeable chemical crosslinker dithiobis (succinimidyl propionate) (DSP) to stabilize the CERT–VAP complex. Then, cell lysate treated with a mild detergent was subjected to co-immunoprecipitation with anti-GFP monoclonal antibody. A larger amount of VAP-A was co-precipitated with mVenus-CERT S315E compared with the CERT WT control (Figure 3B), in agreement with a previous study that showed that the phosphorylated S315-mimetic mutation S315E augments the FFAT motif-dependent binding of CERT to VAP [10]. When cells were exposed to hyperosmotic stress, co-precipitation of CERT WT with VAP-A was augmented coincidently with the elevation of the S315 phosphorylation (Figure 3B). Such augmentation did not occur in the S315A mutant (Figure 3B). For the mVenus-CERT ΔFFAT mutant, while its S315 phosphorylation occurred in response to hyperosmotic stress, its co-precipitation with VAP-A was not detected. These results demonstrate that the augmented binding of CERT to VAP-A in response to hyperosmotic stress depends on the S315 phosphorylation and the FFAT motif in CERT.

We also performed an in situ proximity ligation assay (PLA) to assess the interaction between CERT and VAP-A. PLA detects the physical closeness between two proteins within 40 nm of each other as spot-like signals [12,13]. Although these signals do not necessarily represent the direct interaction between two proteins, the number of spot-like PLA signals can serve as a semi-quantitative measure for assessing the complex formation frequency of the two proteins. Moreover, the subcellular distribution of these signals enables one to evaluate where the interaction between two proteins occurs in cells. In HeLa cells expressing the mVenus-CERT WT, the number of PLA signals per cells increased in response to hyperosmotic stress (Figure 3C), while PLA signals decreased by hyperosmotic stress in the CERT S315A-expressing cells. Hyperosmotic stress did not affect the numbers of the signals in cells expressing CERT ΔFFAT and CERT S315E mutants. These PLA experiment results validated that hyperosmotic stress-induced phosphorylation of S315 enhances the CERT–VAP interaction in cells. 

Notably, the PLA signals representing the CERT–VAP association were distributed throughout the cytoplasm, and the signal distribution patterns did not differ between isosmotic and hyperosmotic conditions. This result raised the possibility that the S315-phosphorylated CERT binds to VAP throughout the ER and not at limited regions at ER–Golgi MCSs. However, the results might be accounted for by another possibility: The Golgi apparatus was fragmented under hyperosmotic conditions, thereby enabling ER–Golgi contact sites to occur throughout the cytoplasm and not to be restricted to the perinuclear regions to which the Golgi is typically localized under normal culture conditions. To examine this possibility, subcellular distributions of VAP-A and the *cis*-Golgi marker GM130, or the *trans*-Golgi network marker TGN46, were analyzed by coimmunostaining. We observed no marked differences in the subcellular distribution of these Golgi markers between the osmotically stressed and the isosmotic control HeLa cells (Figure 3D–G), although hyperosmotic stress was reported to induce fragmentation of the Golgi apparatus in human corneal cells [14]. It should also be noted that hyperosmotic stress increased the perinuclear signals of VAP-A, which were well colocalized with GM130 or TGN46 (Figure 3D,E). The perinuclear redistribution of VAP-A occurred in both the HeLa WT and CERT KO cells, indicating that the VAP-A redistribution is induced in a CERT-independent manner (Figure 3D,E). Sec61β, another ER-resident membrane protein, did not exhibit hyperosmotic treatment-dependent changes in its subcellular distribution (Figure 3F,G). Although it remains unknown how the perinuclear redistribution of VAP-A occurs under hyperosmotic conditions, it may imply an increased association of VAP-A with the perinuclear Golgi complex. Collectively, these results show that the S315 phosphorylated CERT binds to VAP throughout the ER and not at limited regions at the ER–Golgi MCSs. 

### 2.4. Effects of Hyperosmotic Stress on the Synthesis of SM

We next examined effects of hyperosmotic stress on de novo synthesis of SM. Cells were labeled with [^14^C]-serine under isosmotic or hyperosmotic conditions for 1 h, and the metabolically labeled lipids were quantified (Figure 4A,B). Under isosmotic conditions, the patterns of SM synthesis among cell types used were as expected: Compared with the level of SM synthesis in HeLa WT cells, the level in CERT KO was far less. Ectopic expression of mVenus-CERT WT and S315A in CERT KO cells reversed the SM synthesis levels with that in HeLa WT cells, while the ectopic expression of mVenus-CERT ΔFFAT and CERT S315E constructs reversed the levels below and beyond, respectively, the WT control level (Figure 4A). Hyperosmotic stress unexpectedly enhanced the synthesis of SM in CERT KO cells (Figure 4A,B), demonstrating that CERT-independent synthesis of SM is induced under hyperosmotic conditions. In all other cell lines except for the CERT S315E-expressing cell line, hyperosmotic stress enhanced the level of SM synthesis, but this enhancement could be accounted for by the induction of the CERT-independent pathway (Figure 4A). In HeLa CERT KO/mVenus-CERT S315E cells, the SM synthesis was augmented even under isosmotic conditions, and the level was not further enhanced by hyperosmotic stress (Figure 4A). These results suggest that hyperosmotic stress does not upregulate the CERT-dependent synthesis of SM even when CERT–VAP interaction is enhanced by S315 phosphorylation and that hyperosmotic stress somehow renders de novo SM synthesis in a CERT-independent manner.

### 2.5. Hyperosmotic Stress Induces the Preferential Synthesis of the Very-Long-Chain SM via a CERS2-Dependent and CERT-Independent Manner

Scrutinizing the metabolic labeling of TLC images, we noticed that molecular species of de novo synthesized SM were likely altered under hyperosmotic conditions. Metabolic labeling under isosmotic conditions gave typical doublet patterns of SM (Figure 4A), in which the upper bands represent SM with very-long *N*-acyl chain lengths (20 or more carbons), and the lower bands represent SM with long *N*-acyl chain lengths (16 or 18 carbons). Under hyperosmotic conditions, the upper bands of SM were preferentially produced in the metabolic labeling (Figure 4A). Hyperosmotic stress did not affect the doublet patterns of phosphatidylethanolamine (Figure 4A), indicating that the hyperosmotic stress-induced alteration in acyl chain lengths is a sphingolipid-specific event.

Acyl chain lengths of de novo synthesized SM are determined in the step of ceramide production. There are six isozymes in ceramide synthase (CERS1-6) that exhibit different substrate specificity with distinct lengths of acyl-CoA, and CERS2 mainly catalyzes the formation of ceramide species with very-long acyl chains in HeLa cells [15]. To elucidate the contribution of CERS2 to the preferential synthesis of very-long-chain SM (VL-SM) under hyperosmotic conditions, metabolic labeling analysis was performed with CERS2 KO HeLa cells [6]. Long-chain SM (L-SM) was predominantly synthesized in hyperosmotic-stressed CERS2 KO cells as well as in the non-treated cells, and the amounts of synthesized L-SM increased compared with those of parent cells (Figure 4B). This finding is consistent with a previous report that found that in CERS2-deficient cells, the amount of L-ceramides increased to compensate for the shortage of VL-ceramides [6]. This result indicated that the preferential VL-SM synthesis in hyperosmotic environments was dependent on CERS2 and suggested that the VL-ceramides synthesis was upregulated under these conditions. Furthermore, we performed an *in vitro* ceramide synthesis assay to monitor CERS2 activity. After exposure of the parental HeLa mCAT, CERS2 KO, and CERT KO cells to the isotonic control or hyperosmotic conditions, membrane fractions were prepared from the cells. The membrane preparations as the CERS enzyme source were incubated with deuterium-labeled (d7)-sphingosine in the presence of C24:1- or C16:0-acyl CoA, and the quantities of d7-sphingosine-labeled C24:1- and C16:0-ceramides were determined by LC–MS. The membrane fractions from parental HeLa cells and CERT KO cells exhibited a hyperosmotic stimulus-dependent increase in the synthesis of C24:1-ceramide, while no synthesis of C24:1-ceramide was detected in CERS2 KO cell-derived membranes (Figure 4C). On the other hand, none of the parental, CERS2 KO, or CERT KO cell-derived membranes exhibited significant changes in the synthesis of C16:0-ceramide in response to the hyperosmotic stimulus (Figure 4D). These results indicate that hyperosmotic stress selectively upregulates CERS2 activity, which presumably underlies the hyperosmotic stress-induced shift from L-SM to VL-SM, although it remains unclear how CERS2-selective activation occurs under hyperosmotic conditions.

### 2.6. Pharmacologically Induced Merging of the Golgi Apparatus with the ER Does Not Abrogate the Hyperosmotic Stress-Induced Preferential Synthesis of VL-SM

Glucocylceramide (GlcCer) synthase, which catalyzes the conversion of ceramide to GlcCer, has been suggested to be localized to the *cis*/*medial* Golgi regions in mammalian cells [16], and transport of ceramide from the ER to the Golgi site for the synthesis of GlcCer does not require CERT. Because glycosphingolipids have relatively longer acyl chains than SM, it has been proposed that a CERT-independent transport pathway of ceramide for the synthesis of GlcCer preferentially delivers VL-ceramides over l-ceramides [6]. We hypothesized that a VL-ceramides-selective and CERT-independent pathway for delivering ceramide from the ER to the Golgi apparatus might participate in the hyperosmotic stress-induced preferential synthesis of VL-SM even in the absence of CERT. To test this hypothesis, we employed the pharmacological tools Brefeldin A (BFA) and Golgicide A (GCA) (both of which are inhibitors of GBF1: guanine nucleotide exchange factor for ARF [17,18,19]). Because treatment of cells with these drugs induces absorption of the Golgi apparatus into the ER, ceramide is efficiently converted to SM in the drug-treated cells even without any transport pathways to mediate the ER-to-Golgi transport of ceramide. Consistent with previous studies [3,6], when CERT KO cells were treated with BFA or GCA under isosmotic conditions, the cells produced high amounts of both VL-SM and L-SM (Figure 5A). If hyperosmotic stress induces a CERT-independent pathway to deliver ceramide from the ER to the Golgi apparatus, treatment of cells with the ER–Golgi merging drugs should abrogate the preferential synthesis of VL-SM. Nevertheless, the drug treatment of CERT KO cells did not abrogate the preferential synthesis of VL-SM (Figure 5B). These results rule out the possibility that the hyperosmotic stress-induced preferential synthesis of VL-SM is due to a possible upregulation of the putative CERT-independent ceramide transport pathway and further support the notion that the hyperosmotic stress-induced preferential synthesis of VL-SM is due to preferential synthesis of VL-ceramides.

## 3. Discussion

Both the SRM dephosphorylation and S315 phosphorylation occur to activate CERT fully [9,10]. Nevertheless, it remained unknown whether these two phosphorylation events are always induced simultaneously. In this study, we serendipitously found that hyperosmotic stress evokes CERT S315 phosphorylation without affecting the SRM-phosphorylation state. This demonstrates that the two phosphorylation events can be separated depending on the types of stimuli. Under hyperosmotic conditions, the binding of CERT with VAP-A is enhanced in an S315 phosphorylation- and FFAT motif-dependent manner, and this increased binding occurs widely on the ER (Figure 6A), indicating that CERT has the potential to bind VAP anywhere on the ER and that it is not restricted to the ER–Golgi MCSs. When the PH domain of ER-scattering CERT meets PtdIns(4)P of *trans* Golgi regions, which appose in close proximity to subregions of smooth ER [20], CERT might be trapped at the ER–Golgi MCSs.

Although hyperosmotic stress invokes an increase in de novo SM synthesis in HeLa cells, this increase occurs independently of CERT (Figure 4). Thus, the hyperosmotic stress-induced enhancement of CERT binding to the ER is unlikely to enhance CERT’s ER–Golgi ceramide trafficking function. We speculate that the enhanced binding of CERT to the ER is somehow beneficial for hyperosmotic stressed cells. Previous studies demonstrated that aberrant trafficking of ceramide to mitochondria causes cell death [21,22]. Cells undergo shrinkage under hyperosmotic conditions, which might generate otherwise absent emergent physical contacts between the ER and other organelles. 

Moreover, hyperosmotic stress impinges on various types of membrane vesicle trafficking, such as the ER-to-Golgi transports or endosomal pathways [23,24]. These trafficking defects may also disrupt cellular homeostasis, including flows of sphingolipids. Therefore, it would be conceivable that trapping CERT on the ER prevents CERT from mis-sorting ceramide to non-Golgi organelles when abnormal organelle MCSs are generated or lipid metabolic enzymes are not appropriately sorted. 

In addition, we found that hyperosmotic stress induces preferential biosynthesis of VL-SM in a CERT-independent manner (Figure 4A and Figure 5A). The hyperosmotic stress-induced synthesis of VL-SM is not abrogated, even when the ER and Golgi are merged using pharmacological tools (Figure 5B), which eliminated the possibility that the phenomenon is due to an upregulation of a CERT-independent pathway(s) for ER-to-Golgi transport of ceramide. CERS2 is the predominant ceramide synthase (also known as dihydrosphingoine-*N*-acyltransferase) isoform to produce VL-ceramides (more precisely, VL-dihydroceramides) in HeLa cells [6]. In CERS2 KO HeLa cells, L-SM, instead of VL-SM, is predominantly synthesized, and the level of L-SM synthesis is not affected by hyperosmotic stress (Figure 4A). To explain these results together with those of a previous study [6], we propose the model illustrated in Figure 6B. In wild-type HeLa cells under isosmotic conditions, l-ceramides are delivered from the ER to the distal Golgi regions preferentially via the CERT-dependent pathway, whereas VL-ceramides are presumably transported to the proximal or broad Golgi regions via a CERT-independent pathway(s), the entity of which remains elusive. Under hyperosmotic conditions, the synthesis of VL-ceramides is enhanced, and therefore, a greater amount of VL-ceramides is delivered to the Golgi via the CERT-independent pathway, thereby enhancing the synthesis of VL-SM. This model would account for that hyperosmotic stress inducing the synthesis of VL-SM even in the absence of CERT and also would explain that hyperosmotic stress does not enhance L-SM synthesis in the absence of CERS2. More studies will be needed to address unresolved questions—for example, how does hyperosmotic stress enhance the CERS2-dependent synthesis of VL-ceramides, and what is the entity of the CERT-independent pathway?

The osmolarity of the medium supplemented with 800 mM sorbitol was about 1100 mOsm. The renal inner medulla is exposed to such severe hyperosmotic environments [25]. SM molecular species with different acyl-chain lengths were shown to be distributed among different subregions of the kidney [26]. It could be possible that the hyperosmotic stress-induced phosphorylation of CERT S315 partly underlies the region-dependent distribution of various SM molecular species in the kidney.

Although the biological importance of VL-SM under osmotic stress has not yet been elucidated, a previous study showed that an increase in VL-sphingolipids in yeast cells may strengthen membrane integrity and may endow some tolerance to hyperosmotic stress [27]. By analogy, the increased VL-SM synthesis in mammalian cells might be an acute response for adapting to hyperosmotic conditions. Considering that ceramides act as modulators of various proteins [28], the shift from l-ceramides to VL-ceramides might affect cellular signaling events. For example, C16-ceramide, but not other ceramide species, stabilizes the p53 tumor suppressor, a key regulator in various fundamental cellular events such as the cell cycle, apoptosis, and survival in response to diverse stimuli [29]. On the other hand, acute hyperosmotic stress induces caspase-mediated apoptosis [30]. Thus, it is conceivable that a shift from l-ceramides to VL-ceramides may destabilize p53 and consequently attenuate p53-mediated hyperosmotic stress-induced cell death. Further studies will be needed to address these possibilities.

In conclusion, hyperosmotic stress evokes CERT S315 phosphorylation without affecting the SRM-phosphorylation state, showing compelling evidence that the two phosphorylation events can be separated. Under hyperosmotic conditions, the binding of CERT with VAP-A is enhanced in an S315 phosphorylation- and FFAT motif-dependent manner, and this increased binding occurs widely on the ER. This reveals that CERT has the potential to bind VAP anywhere on the ER and that it is not restricted to the ER–Golgi MCSs.

## 4. Materials and Methods

### 4.1. HeLa Cell Lines and Cell Culture

The HeLa-mCAT#8 cell line that stably expresses the mouse ecotropic retroviral receptor mCAT-1 was used as the HeLa WT cell line, and its *CERT1*-disrupted mutant cell line was used as the HeLa CERT KO cell line [5] unless otherwise noted. To avoid unwanted overexpression of CERT cDNAs, the HeLa CERT KO/shCERT cell line was used. The HeLa CERT KO/shCERT cell line was established as described in a separate publication (Goto et al., submitted). In brief, the CERT KO cells were transfected using a retroviral vector containing a short hairpin RNA against *CERT1* RNA to interfere with the expression of *CERT1* mRNA. Then, a stable clone was isolated and used as the parental HeLa CERT KO/shCERT cell line. The CERS2 KO cell line (#16) was established as described previously [6]. HeLa cells were cultured in high glucose Dulbecco’s modified Eagle medium (D-MEM; 044-29765, Wako, Osaka, Japan) supplemented with 10% heat-inactivated fetal calf serum.

### 4.2. Plasmids

cDNA fragments encoding the full-length human CERT wild-type and its mutants were digested from previously described vectors [4,9] and subcloned into pMX-IB vectors containing the human influenza HA tag or the mVenus tag. 

### 4.3. Retroviral Transfection and Establishment of Stable Cell Lines

Retroviral transfection of HeLa cells with various cDNA constructs was performed using the pMXs-IB-based retroviral vector as described previously [5]. After selecting transfected cells with 7.5 µg/mL blasticidin (#029-18701, Wako), cell clones were purified by limited dilution. Cell cloning was not performed for blasticidin-selected cells expressing mVenus-tagged CERT; instead, a population of cells expressing the fluorescent protein in a limited range was selected by FACS. After propagation, the FACS-selected cells were used for analysis.

### 4.4. Western Blotting

Cells were seeded under subconfluent conditions and were lysed in the lysis buffer (50 mM Tris-HCl (pH 7.4), 1 mM EDTA, 1 mM EGTA, 100 mM NaCl, 50 mM NaF, 5 mM sodium pyrophosphate, 10 mM disodium-β-glycerophosphate, cOmplete™ Protease Inhibitor Cocktail (#11836153001, Roche, Basel, Switzerland), 1% Phosphatase Inhibitor Cocktail 2 (P5726, Sigma-Aldrich, Burlington, MA, USA), and 1% Phosphatase Inhibitor Cocktail 3 (P0044, Sigma-Aldrich)) containing 1% Triton X-100. After centrifugation (14,000× rpm for 10 min at 4 °C), the supernatants of the lysates were collected, and the protein concentrations were quantified with the bicinchoninic acid (BCA) method (#23227, Thermo Scientific, Waltham, MA, USA). Equal volumes of the immunoprecipitated fraction or equal amounts of the total proteins in the input fraction were separated by electrophoresis on sodium dodecyl sulfate (SDS)/7.5% or 15% polyacrylamide gels and transferred to polyvinylidene difluoride (PVDF) membrane (#1620177, Bio-Rad, Hercules, CA, USA). Precision Plus Protein™ All Blue Prestained Protein Standards (#1610373, Bio-Rad) were used as the molecular mass standards. After blocking, the membranes were immunoblotted using rabbit polyclonal anti-CERT antibody (ab72536, Abcam, Cambridge, UK), monoclonal anti-HA antibody (clone 3F10) conjugated with horseradish peroxidase (HRP) (#12013819001, Roche, Basel, Switzerland), mouse monoclonal anti-GAPDH antibody (clone 5A12, #016-25523, Wako), rabbit polyclonal anti-phospho-Serine315 CERT antibody generated by immunization [10], rat monoclonal anti-GFP antibody (clone GF090R, #04404-26, Nacalai, Kyoto, Japan), rabbit polyclonal anti-VAP-A antibody (HPA009174, Sigma-Aldrich), and rabbit polyclonal anti-CERT *C*-terminus antiserum generated by immunization [9]. The designated proteins were visualized with the HRP-conjugated secondary antibodies (Jackson ImmunoResearch, West Grove, PA, USA) and SuperSignal™ West Femto Maximum Sensitivity Substrate (#34095, Thermo Scientific). The signals were captured using the LuminoGraph image analyzer (ATTO, Tokyo, Japan), and the images were processed with Fiji/ImageJ software (NIH). The relative levels of S315 phosphorylation were presented as the ratios of the band intensities normalized to those of the immunoprecipitated HA-CERT. The sum of the ratios in each condition was normalized to 3 (Figure 1E,G), 16 (Figure 1F), or 5 (Figure 2A) (the number of conditions; arbitrary unit) in each experiment. The unprocessed images of the blots are shown in Appendix A.

### 4.5. Protein Phosphatase Treatment

Cells were lysed with the lysis buffer without the phosphatase inhibitor cocktails, and the lysates (40 µL) were incubated with 800 units of λPPase (P0753, New England Biolabs, Ipswich, MA, USA) at 30 °C for 30 min.

### 4.6. Immunoprecipitation

Immunoprecipitation was performed as previously described [10] with minor modifications. Briefly, cells were seeded in a 6-well plate under subconfluent conditions and were lysed in the lysis buffer containing 1% Triton X-100. After centrifugation, the supernatants were collected as the input fraction, and equal amounts of proteins were incubated with anti-HA antibody-conjugated agarose (A2095, Merck Millipore, Burlington, MA, USA) or anti-GFP antibody-conjugated agarose (#060830-05, Nacalai) for 1 h at 4 °C with gentle shaking. The resins were washed twice with the lysis buffer containing 0.1% Triton X-100 and were incubated with 1 × NuPAGE^®^ lithium dodecyl sulfate sample buffer with 50 mM dithiothreitol at 70 °C for 5 min. After centrifugation, the supernatants were collected and subjected to Western blotting as the immunoprecipitated fraction.

### 4.7. Immunocytochemistry

Cells were seeded on cover glasses (#12-545-80, Fisherbrand, Waltham, MA, USA) that were coated with collagen solution (TMTCC-050, TOYOBO, Tokyo, Japan) according to the manufacturer’s protocol. After 800 mM sorbitol treatment for 60 min at 37 °C, cells were fixed with Mildform 10N (#133-10311, Wako) for 15 min at room temperature. The cells were permeabilized with 0.1% Triton X-100 in phosphate-buffered saline (PBS) for 10 min and incubated with a blocking solution containing 3% BSA in PBS for 30 min. The cells were incubated with the following primary antibodies: rat monoclonal anti-HA (#1867423001, Roche), rabbit polyclonal anti-VAP-A (HPA009174, Sigma-Aldrich), rabbit polyclonal anti-Sec61β (#07-205, Sigma-Aldrich), mouse monoclonal anti-GM130 (#610823, BD Biosciences, San Jose, CA, USA), or sheep polyclonal anti-TGN46 (AHP500, Bio-Rad). Then, the cells were incubated with the following secondary antibodies: Alexa Fluor 488-conjugated goat anti-rabbit (A11034, Invitrogen, Waltham, MA, USA), Alexa Fluor 594-conjugated goat anti-mouse (A11037, Invitrogen), or Alexa Fluor 594-conjugated donkey anti-sheep (#713-586-147, Jackson ImmunoResearch). The cover glasses were mounted on a slide glass (S1214, MATSUNAMI) using Fluoromount reagent (K024, Diagnostic BioSystems, Pleasanton, CA, USA). The cells were visualized using a wide-field fluorescence microscope, BioZero (BZ-X710, Keyence, Tokyo, Japan), equipped with a Plan Apo V 60 × 1.40 oil immersion objective. Haze reduction function (condition 3), which applies a no-neighbor deconvolution algorithm to the captured image, was used to eliminate fluorescence blurring caused by scattered light and to capture clear images with high contrast. The acquired images were processed with Fiji/ImageJ software (NIH).

### 4.8. In situ Proximity Ligation Assay (PLA)

Cells were seeded on the cover glasses coated with collagen solution and processed as described in Section 4.7. permeabilization step. PLA was carried out using Duolink In situ Detection Reagents Red (DUO92008, Sigma-Aldrich) according to the manufacturer’s protocol. The cells were incubated with rat monoclonal anti-GFP antibody (#04404-26, Nacalai) and rabbit polyclonal anti-VAP-A antibody, followed by PLA probe anti-rat PLUS and PLA probe anti-rabbit MINUS (DUO92005, Sigma-Aldrich). The PLA probe anti-rat PLUS was prepared using Duolink In situ Probemaker PLUS (DUO92009, Sigma-Aldrich) against donkey anti-rat antibody (#712-005-150, Jackson ImmunoResearch) according to the manufacturer’s protocol. The cover glasses were mounted on a slide glass using Duolink In situ Mounting Medium with DAPI (DUO82040, Sigma-Aldrich), and the cells were analyzed by a BioZero digital microscope (Haze reduction function: condition 3) equipped with a Plan Apo V 60 × 1.40 oil immersion objective. The fluorescence and bright images were processed by Fiji/ImageJ software to measure the number of spots of PLA signals per cells.

### 4.9. Metabolic Labeling of Lipids with Radioactive Serine

Metabolic labeling of lipids with radioactive serine was performed as described previously [5], except that cells were cultured in the normal culture medium overnight at 37 °C to subconfluence in a 12-well plate and then incubated in a serum-free DMEM containing 1.5 µCi of L-[U-^14^C]-serine (MC-265, Moravek Inc., Brea, CA, USA) for 1 h under various osmotic conditions. The relative values of each lipid are presented as the ratios of the band intensities normalized to the amounts of proteins used in each condition. The sum of the ratios in each condition was normalized to 12 (Figure 4A) or 6 (Figure 4B) (the number of conditions; arbitrary unit) in each experiment. The unprocessed TLC images are shown in Appendix A.

### 4.10. In Vitro Ceramide Synthesis Assay

HeLa cells were suspended in cell suspension buffer (50 mM HEPES-NaOH (pH 7.4), 150 mM NaCl, 10% glycerol, 1 mM DTT, 1% Phosphatase Inhibitor Cocktail 2 (Sigma-Aldrich), 1% Phosphatase Inhibitor Cocktail 3 (Sigma-Aldrich), and cOmplete™ Protease Inhibitor Cocktail (Roche)) and then lysed by sonication. After removing the cell debris from the lysate by centrifugation (300× *g* for 5 min at 4 °C), the supernatant fraction was centrifuged (100,000× *g* for 30 min at 4 °C). The obtained participant, referred to as the membrane fraction, was suspended in cell suspension buffer and used as the enzyme source for the in vitro ceramide synthesis assay as follows. The membrane fraction (40–50 µg protein) was incubated with 5 µM deuterium-labeled (d7)-sphingosine (#860657P, Avanti Polar Lipids, Alabaster, AL, USA) and 25 µM C24:1- acyl CoA (#870725P, Avanti Polar Lipids) or C16:0-acyl CoA (P9716, Sigma-Aldrich) in 100 µL of the reaction buffer (the cell suspension buffer containing 2 mM MgCl2 and 0.1% digitonin) for 30 min at 37 °C. Following the addition of 1 nmol of C17 ceramide (#860517P, Avanti Polar Lipids) to the reaction mixture as an internal standard for LC–MS/MS, lipids were extracted using Bligh and Dyer’s method [31], and the organic phase retrieved was evaporated and redissolved in 130 µL of methanol. Then, an aliquoted sample was subjected to an LC–MS system comprising a Prominence UFLC system (Shimadzu, Tokyo, Japan) coupled to a 3200 QTRAP System (SCIEX, Framingham, MA, USA), as previously described [32]. The levels of declustering potential, entrance potential, and collision energy for deuterated ceramides were optimized for each target. The multiple-reaction monitoring (MRM) mode was used to measure the deuterated ceramides. The ion pairs of d7-d18:1/C16:0 ceramide and d7-d18:1/C24:1 ceramide were *m/z* = 545.6–271.4 and 655.6–271.4, respectively. Each ion pair of the molecular species in MRM was monitored for 30 ms with a resolution of unit. The contents of individual ceramide were calculated by the peak area of analyte to the peak area of the internal standard. Data acquisition and analysis were performed using Analyst Software version 1.6. (SCIEX). The sum of the signals of each condition was normalized to 6 (the number of conditions; arbitrary unit) in each experiment.

### 4.11. Statistical Analysis

Statistical analyses were carried out using R software. The Student’s *t*-test was used for two-group comparisons, setting *p* < 0.05 as a statistical significance criterion.

## Figures and Tables

**Figure 1 ijms-23-04025-f001:**
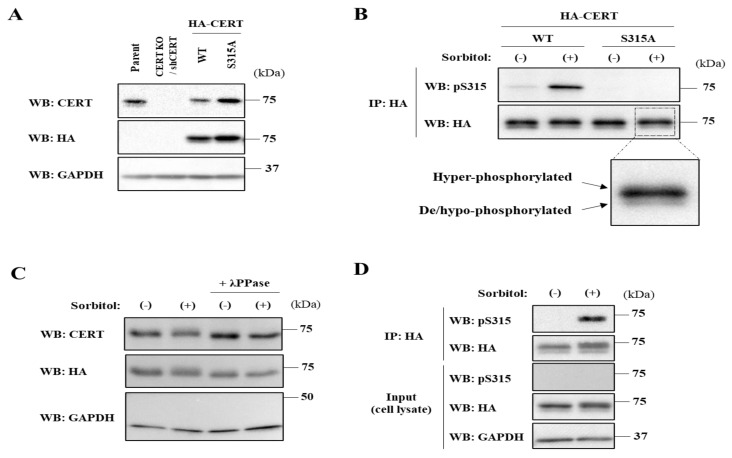
Hyperosmotic stress induces the phosphorylation of CERT S315 residue without changing the SRM-phosphorylation state. (**A**) The established cell lines expressing HA-CERT wild-type (WT) or S315A mutant with near endogenous level. The cells were lysed and immunoblotted with the respective antibodies against CERT, HA-epitope, and the loading control GAPDH. (**B**) The S315 phosphorylation of CERT was monitored using a phosphorylated S315 (S315p)-specific antibody. CERT KO/shCERT HeLa cells expressing HA-CERT WT or S315A mutant with near endogenous level were treated with 600 mM sorbitol for 1 h. The lysates were subjected to immunoprecipitation with HA-agarose, and the eluates were immunoblotted with the respective antibody against S315p and HA. The magnified area shows the CERT doublet due to its phosphorylation states of the SRM. Two independent experiments were carried out, and similar results were obtained. (**C**) Protein phosphatase treatment shifts the upper band of the CERT doublet to the lower band independent of whether there is or is not hyperosmotic stress. CERT KO/shCERT HeLa cells expressing HA-CERT WT with near endogenous level were treated with 800 mM sorbitol for 1 h, and the lysates were incubated with the protein phosphatase (λPPase). Two independent experiments were carried out and similar results were obtained. (**D**) CERT KO/shCERT HeLa cells expressing HA-CERT WT with near endogenous level were treated with 800 mM sorbitol for 1 h. The lysates and the eluate of immunoprecipitated samples were immunoblotted with the respective antibodies. Two independent experiments were carried out and similar results were obtained. (**E**) Hyperosmotic stress induces the S315 phosphorylation without changing the phosphorylation state of the SRM. CERT KO/shCERT HeLa cells expressing HA-CERT WT with near endogenous level were treated with 800 mM sorbitol for 1 h or 3 µM ISP-1 for 24 h. The lysates were immunoprecipitated with HA-agarose, and the samples were immunoblotted with the respective antibody against S315p and GAPDH. The relative levels of S315 phosphorylation were evaluated as described in Section 4. The bars represent the means (n = 3), and the white circles indicate the values from each experiment. (**F**) Osmolarity and time dependence of the CERT S315 phosphorylation. CERT KO/shCERT HeLa cells expressing HA-CERT WT with near endogenous level were treated with various concentrations of sorbitol for various time periods. The eluates of immunoprecipitated samples and the total cell lysates were immunoblotted with the indicated antibodies. The bars represent the means (n = 3), and the white circles indicate the values from each experiment. (**G**) Ionic osmolyte also induces the CERT S315 phosphorylation. CERT KO/shCERT HeLa cells expressing HA-CERT WT with near endogenous level were treated with 300 mM NaCl or 600 mM sorbitol for 1 h. The eluates of immunoprecipitated samples and the total cell lysates were immunoblotted with the indicated antibodies. The bars represent the means (n = 3), and the white circles indicate the values from each experiment.

**Figure 2 ijms-23-04025-f002:**
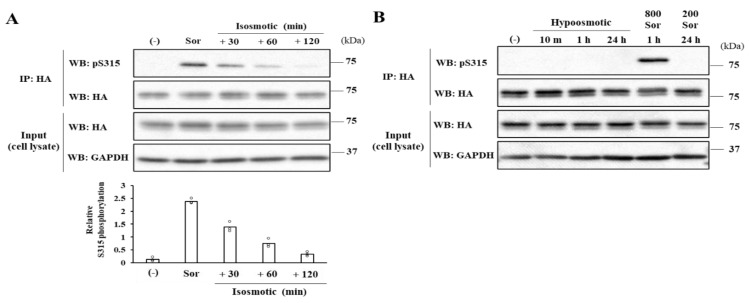
Effects of various osmotic stresses on the CERT S315 phosphorylation. (**A**) The osmotic stress-induced elevations of the S315 phosphorylation are reversibly decreased. After the incubation in a medium containing 800 mM sorbitol for 1 h, CERT KO/shCERT HeLa cells expressing HA-CERT WT with near endogenous level were further incubated in an isosmotic medium for the indicated times. The eluates of the immunoprecipitated samples and the total cell lysates were immunoblotted with the indicated antibodies. The bars represent the means (n = 3), and the white circles indicate the values from each experiment. (**B**) Hypoosmotic or mild but prolonged hyperosmotic stresses do not augment the S315 phosphorylation. For hypoosmotic stress, CERT KO/shCERT HeLa cells expressing HA-CERT WT with near endogenous level were incubated in a medium diluted with distilled water by two-thirds for the indicated times. For relatively mild (500 mOsm) but prolonged hyperosmotic stress, cells were treated with 200 mM sorbitol for 24 h. The eluates of the immunoprecipitated samples and the total cell lysates were immunoblotted with the indicated antibodies. Two independent experiments were carried out, and similar results were obtained.

**Figure 3 ijms-23-04025-f003:**
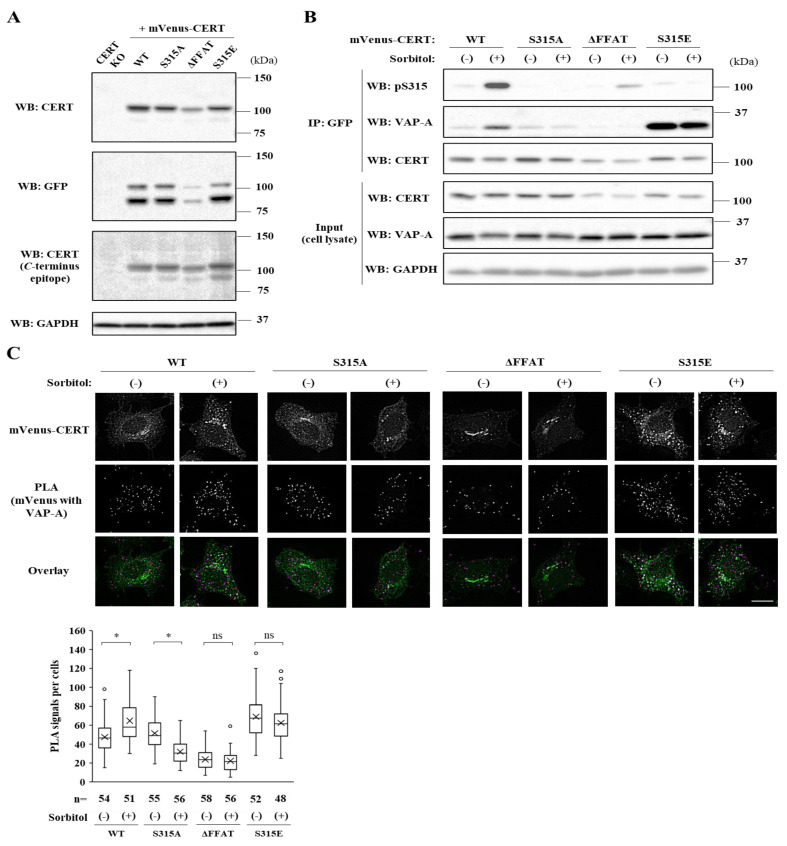
The hyperosmotic stress-induced CERT S315 phosphorylation augments the binding to VAP-A. (**A**) The cell lines ectopically expressing each mVenus-CERT mutant were established from CERT KO cells as parents. The lysates were immunoblotted with the respective antibody (CERT, GFP, and GAPDH) or the antiserum against CERT (*C*-terminus epitope). (**B**) The enhanced binding to VAP-A occurs in the S315 phosphorylation-dependent manner. An increase in co-immunoprecipitated VAP-A with CERT WT occurs along with the S315 phosphorylation, but the increased binding to VAP-A in cells expressing CERT S315A is not observed. Cells expressing each mVenus-CERT variant were treated with 800 mM sorbitol for 1 h. Then, cells were treated with 2 mM crosslinker DSP on ice for 30 min. The lysates were subjected to immunoprecipitation with GFP-agarose. The eluates or the lysates were immunoblotted with the respective antibody against S315p CERT, VAP-A, CERT, and GAPDH. Two independent experiments were performed, and similar results were obtained. (**C**) The physical proximity between CERT and VAP-A under hyperosmotic conditions. Cells expressing each mVenus-CERT variant were treated with 800 mM sorbitol for 1 h and subjected to the proximity ligation assay (PLA). The representative fluorescence micrographs (in overlay image, green: mVenus-CERT, magenta: PLA signals) are shown. The number of PLA dot signals per cells was counted using Fiji/ImageJ software (NIH) for the indicated number of cells. The box plot demonstrates the number of PLA signals per cells under the respective condition in a representative experiment. The line and cross in the box plot indicate the median and mean, respectively. Scale bars: 10 µm. Two independent experiments were carried out, and similar results were obtained. *: *p* < 0.05, ns: not significant (Student’s *t*-test). (**D**–**G**) Effects of hyperosmotic stress on the morphology of the ER and the Golgi apparatus. Cells were treated with 800 mM sorbitol for 1 h, and immunocytochemistry was performed. The representative fluorescence micrographs are shown in the overlay images—green: (**D**,**E**) VAP-A or (**F**,**G**) Sec61β and magenta: (**D**,**F**) GM130 or (**E**,**G**) TGN46. Scale bars: 10 µm. Two independent experiments were carried out, and similar results were obtained.

**Figure 4 ijms-23-04025-f004:**
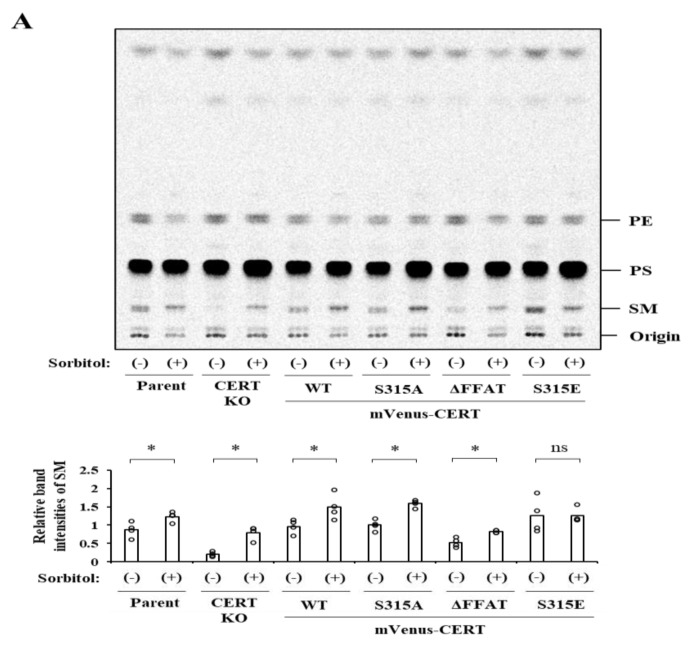
Effects of hyperosmotic stress on SM biosynthesis. (**A**) HeLa CERT KO cell lines stably expressing various mVenus-CERT variants were labeled with [^14^C]-serine/800 mM sorbitol-containing medium or isotonic medium for 1 h. Lipids extracted from the cells were separated by TLC using the buffer (methyl acetate/1-propanol/chloroform/methanol/0.25% KCl—50/50/50/20/18, vol/vol). A representative radioactive image of the analyzed TLC plates is shown. Relative band intensities of SM were calculated as described in Section 4. White bars in the graph represent the mean (n = 4), and the white circles indicate the values from each experiment. *: *p* < 0.05 (Student’s *t*-test). PtdEtn, phosphatidylethanolamine; PtdSer, phosphatidylserine; SM, sphingomyelin. (**B**) Metabolic labeling analysis was performed in CERS2 KO cells as described in (A). The upper panel shows the whole TLC image. The lower panel shows the vertically magnified image of SM of the upper panel to discern the doublet patterns. The open and filled bars indicate the relative band intensities of very-long-chain SM (VL-SM) and long-chain SM (L-SM), respectively. Data are presented as the mean ± SD (n = 3). (**C**,**D**) Effects of hyperosmotic stress on CERS2 activity. Following exposure of the indicated cells to the isotonic control or hyperosmotic conditions, a membrane fraction was prepared from the cells. The membrane fraction was incubated with d7-sphingosine in the presence of C24:1-acyl CoA or C16:0-acyl CoA for 30 min at 37 °C. Then, the quantities of d7-sphingosine-labeled C24:1-ceramide (**C**) and C16:0-ceramide (**D**) synthesized were determined by LC–MS. The relative levels of synthesis of d7-sphingosine-labeled ceramides were calculated as described in Section 4. Data are presented as the means (n = 3), and the white circles indicate the values from each experiment. *: *p* < 0.05 (Student’s *t*-test).

**Figure 5 ijms-23-04025-f005:**
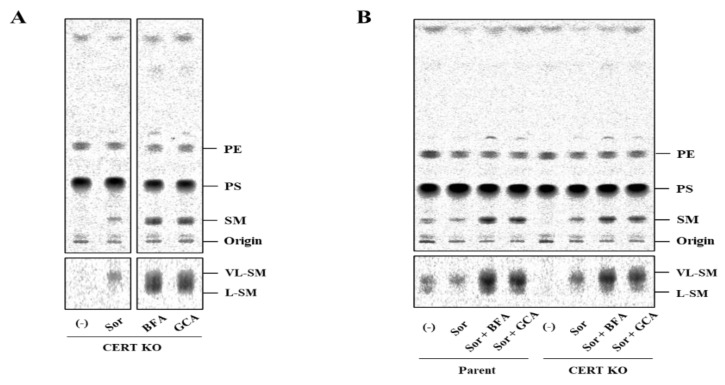
Effect of pharmacological merging of the Golgi apparatus with the ER on hyperosmotic stress-evoked VL-SM synthesis. (**A**) CERT KO cells were labeled with [^14^C]-serine/800 mM sorbitol, 2.5 µM BFA, 20 µM GCA-containing medium, or normal culture medium for 1 h. Metabolically labeled lipids were analyzed by TLC. The representative cropped TLC images on the same plate (the upper panels), and vertically magnified images of the SM (the lower panels) to discern the doublet, are aligned. Two independent experiments were carried out, and similar results were obtained. (**B**) Cells were labeled with [^14^C]-serine in an 800 mM sorbitol-containing medium in the presence of 2.5 µM BFA or 20 µM GCA. Metabolically labeled lipids were analyzed by TLC. The representative whole TLC image (the upper panel) and magnified image of the SM doublets (the lower panel) are shown. Two independent experiments were carried out, and similar results were obtained.

**Figure 6 ijms-23-04025-f006:**
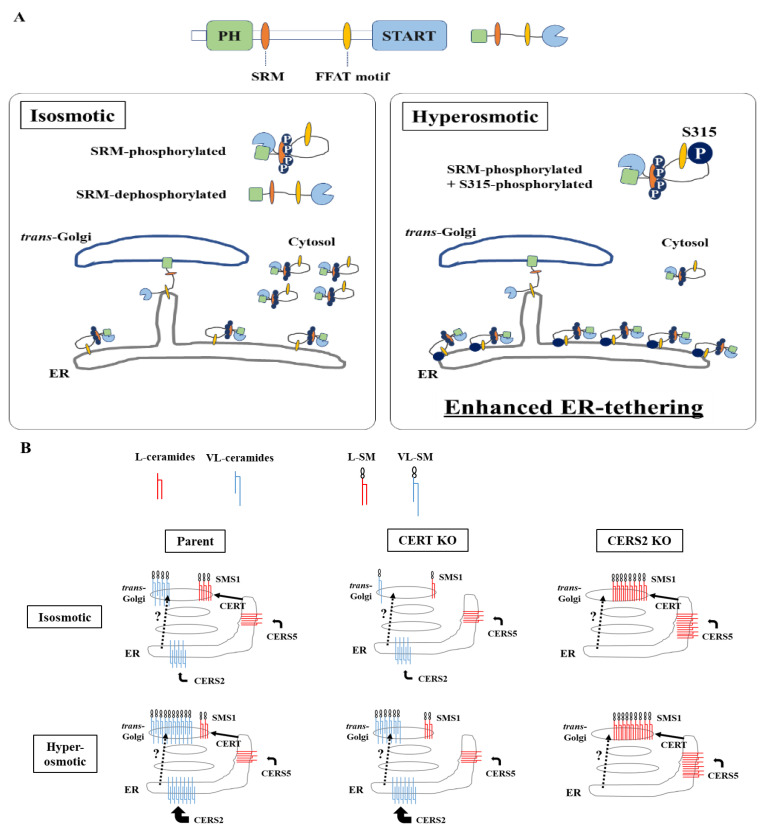
Schematic diagrams of CERT and the ceramide metabolism in response to hyperosmotic stress. (**A**) The enhanced ER-tethering of CERT via the S315-phosphorylation under hyperosmotic conditions. Under isosmotic conditions, a major population of CERT, multiply phosphorylated in the SRM, localizes to the ER or the cytosol, while a minor population of CERT, which is de/hypo-phosphorylated in the SRM, may localize at the ER–Golgi MCSs. Under hyperosmotic conditions, a majority of CERT are phosphorylated at the S315 residue with SRM-hyperphosphorylation resulting in enhancement of tethering to the ER, rather than localization at the ER–Golgi MCSs, via the increased binding of CERT to VAP-A. (**B**) A hypothetical model that may account for the increased VL-SM synthesis under hyperosmotic conditions in a CERT-independent manner. Ceramide synthesized in the ER is transported to the *trans*-Golgi, where SM synthase 1 (SMS1) locates, via a CERT-dependent (solid line) or -independent (dotted line) pathway. In wild-type HeLa cells under isosmotic conditions, long-chain ceramides (L-ceramides), which are mainly catalyzed by CERS5, are delivered from the ER to the distal Golgi regions preferentially via the CERT-dependent pathway, whereas very-long-chain ceramides (VL-ceramides), which are mainly catalyzed by CERS2, are presumably transported to the proximal or more broad Golgi regions via a CERT-independent pathway(s), of which the entity remains elusive. Under hyperosmotic conditions, the synthesis of VL-ceramides is enhanced, and therefore, a greater amount of VL-ceramides is delivered to the Golgi via the CERT-independent pathway, thereby enhancing the synthesis of VL-SM.

## Data Availability

Unprocessed images of the immunofluorescence and raw data for quantitative results have been posted at the public data repository figshare (https://doi.org/10.6084/m9.figshare.17030093).

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
