# Peer review of "Hyperosmotic Stress Induces Phosphorylation of CERT and Enhances Its Tethering throughout the Endoplasmic Reticulum"

_ijms, 2022, doi:10.3390/ijms23074025_

Round 1
Reviewer 1 Report
The manuscript titled "Hyperosmotic Stress Induces Phosphorylation of CERT and Enhances Its Tethering Throughout the Endoplasmic Reticulum" demonstrated that 800 mM sorbitol treatment to HeLa cells increased S315-specific phosphorylation levels in HA-CERT and did not affect SRM phosphorylation status. The authors argued that the elevated S315 phosphorylation on overexpressed CERT enhanced its interaction with VAP-A and its proximity to VAP-A, and that the interaction was not limited to the perinuclear regions. Also, the authors demonstrated that VAP-A redistributed to perinuclear regions/proximal to Golgi markers under sorbitol treatment, although the localization patterns of Golgi markers were largely unchanged. The authors showed that the sorbitol treatment increased the synthesis of very-long-chain sphingomyelin, which was dependent on CERS2 expression.
The research was conducted very soundly and the manuscript is well-written. The presentation and the interpretation of data were accurate and precise. I acknowledge the authors' scientific integrity and honesty that can be seen in the raw gel images in the supplementary data.
I have minor comments.
- It is solid that overexpressed CERT phosphorylation status changes by hyperosmotic stress, but it is still not presented that endogenous CERT behaves as the overexpressed CERT does. Please provide an immunoblot that can monitor the phosphorylation upon osmotic stress.
- Although CerS2 is well-known to have a clear biochemical substrate preference toward VL-fatty acids, the increase of VL-SM synthesis may not be attributed to CerS2 until its activity is shown to change by osmotic stress. Please provide an experimental result to show the changes in CerS2 activity upon osmotic stress.
- Please provide a discussion that could explain the biological importance of VL-SM under osmotic stress.
Author Response
Response to Reviewer 1 Comments
Point 1: It is solid that overexpressed CERT phosphorylation status changes by hyperosmotic stress, but it is still not presented that endogenous CERT behaves as the overexpressed CERT does. Please provide an immunoblot that can monitor the phosphorylation upon osmotic stress.
Response 1: Thank you for your important comment. Unfortunately, we could not obtain appropriate data for technical reasons. As shown in the new Figure 1D in our revised manuscript, it is necessary to concentrate CERT from the cell lysate fraction by immunoprecipitation (we usually perform this with an anti-HA-epitope tag monoclonal antibody), to detect the S315 phosphorylated form by western blotting analysis with anti-CERT S315p polyclonal antibodies. Thus, to address your question, we attempted to immunoprecipitate the endogenous CERT using commercially available goat anti-CERT polyclonal antibodies and found that these anti-CERT antibodies did not work for immunoprecipitation. We have briefly described this technical issue in the revised manuscript as follows. “To detect the S315 phosphorylated form by western blotting analysis with anti-CERT S315p polyclonal antibodies, we had to concentrate CERT from the cell lysate fraction by immunoprecipitation (Figure 1D). HA-CERT was immunoprecipitated with an anti-HA-epitope monoclonal antibody, but not with commercially available anti-CERT polyclonal antibodies. Thus, for this technical reason, we could not verify that the hyperosmotic stress affected the level of phosphorylation of S315 of the endogenous CERT.” (Lines 107-113)
As shown in both the original and revised manuscripts, HeLa CERT KO/shCERT cells expressed HA-CERT at near endogenous levels, not in an overexpressed manner (Figure. 1A). We believe that the S315 phosphorylation responses to hyperosmotic stress in this cell line recapitulate the responses of endogenous CERT.
Figure 1D in the revised version
Point 2: Although CerS2 is well-known to have a clear biochemical substrate preference toward VL-fatty acids, the increase of VL-SM synthesis may not be attributed to CerS2 until its activity is shown to change by osmotic stress. Please provide an experimental result to show the changes in CerS2 activity upon osmotic stress.
Response 2: To address this comment, we performed an in vitro ceramide synthesis assay using the membrane fractions prepared from hyperosmotic-stressed cells. The membrane fractions were incubated with deuterium-labeled (d7)-sphingosine in the presence of C24:1- or C16:0- acyl CoA, and the quantities of synthesized C24:1 or C16:0-ceramide with d7-sphingosine were determined by LC–MS. As shown in the new Figures 4C and D, hyperosmotic-treatment resulted in the increased synthesis of C24:1-ceramide (Figure 4C), whereas there was no increase in C16:0-ceramide synthesis (Figure 4D). These results showed that hyperosmotic stress selectively upregulates the activity of CERS2 and that increased VL-SM synthesis is partly attributable to the increased VL-ceramides. We have described these results in the revised manuscript as follows.
“Furthermore, we performed an in vitro ceramide synthesis assay to monitor CERS2 activity. After exposure of the parental HeLa mCAT, CERS2 KO, and CERT KO cells to the isotonic control or hyperosmotic conditions, membrane fractions were prepared from the cells. The membrane preparations as the CERS enzyme source were incubated with deuterium-labeled (d7)-sphingosine in the presence of C24:1- or C16:0-acyl CoA, and the quantities of d7-sphingosine-labeled C24:1- and C16:0-ceramides were determined by LC–MS. The membrane fractions from parental HeLa cells and CERT KO cells exhibited a hyperosmotic stimulus-dependent increase in the synthesis of C24:1-ceramide, while no synthesis of C24:1-ceramide was detected in CERS2 KO cell-derived membranes (Figure 4C). On the other hand, none of the parental, CERS2 KO, or CERT KO cell-derived membranes exhibited significant changes in the synthesis of C16:0-ceramide in response to the hyperosmotic stimulus (Figure 4D). These results indicated that hyperosmotic stress selectively upregulates CERS2 activity, which presumably underlies the hyperosmotic stress-induced shift from L-SM to VL-SM, although it remains unclear how CERS2-selective activation occurs under hyperosmotic conditions.” (Lines 361-375)
Figures 4C and D in the revised version are also shown below.
For the new data shown above, we have also added new paragraphs to the Materials and Method section and updated the legend to Figure 4 as follows.
In vitro ceramide synthesis assay
HeLa cells were suspended in cell suspension buffer [50 mM HEPES-NaOH (pH 7.4), 150 mM NaCl, 10% glycerol, 1 mM DTT, 1% Phosphatase Inhibitor Cocktail 2 (Sigma-Aldrich), 1% Phosphatase Inhibitor Cocktail 3 (Sigma-Aldrich), and cOmplete™ Protease Inhibitor Cocktail (Roche)] and then lysed by sonication. After removing the cell debris from the lysate by centrifugation (300×g for 5 min at 4℃), the supernatant fraction was centrifuged (100,000×g for 30 min at 4℃). The obtained participant, referred to as the membrane fraction, was suspended in cell suspension buffer and used as the enzyme source for the in vitro ceramide synthesis assay as follows. The membrane fraction (40–50 µg protein) was incubated with 5 µM deuterium-labeled (d7)-sphingosine (#860657P, Avanti Polar Lipids) and 25 µM C24:1- acyl CoA (#870725P, Avanti Polar Lipids) or C16:0-acyl CoA (P9716, Sigma-Aldrich) in 100 µl of the reaction buffer (the cell suspension buffer containing 2 mM MgCl2 and 0.1% digitonin) for 30 min at 37℃. Following the addition of 1 nmol of C17 ceramide (#860517P, Avanti Polar Lipids) to the reaction mixture as an internal standard for LC–MS/MS, lipids were extracted using Bligh and Dyer’s method (Bligh and Dyer, Can. J. Biochem. Physiol., 1959), and the organic phase retrieved was evaporated and redissolved in 130 µl of methanol. Then, an aliquoted sample was subjected to an LC–MS system comprising a Prominence UFLC system (Shimadzu) coupled to a 3200 QTRAP System (SCIEX), as previously described (Nakao et al, Comm. Chem., 2019). The levels of declustering potential, entrance potential, and collision energy for deuterated ceramides were optimized for each target. The multiple-reaction monitoring (MRM) mode was used to measure the deuterated ceramides. The ion pairs of d7-d18:1/C16:0 ceramide and d7-d18:1/C24:1 ceramide were m/z=545.6–271.4 and 655.6–271.4, respectively. Each ion pair of the molecular species in MRM was monitored for 30 ms with a resolution of unit. The contents of individual ceramide were calculated by the peak area of analyte to the peak area of the internal standard. Data acquisition and analysis were performed using Analyst Software version 1.6. (SCIEX). The sum of the signals of each condition was normalized to 6 (the number of conditions; arbitrary unit) in each experiment.
For the legend to Figure 4:
(C, D) Effects of hyperosmotic stress on CERS2 activity. Following exposure of the indicated cells to the isotonic control or hyperosmotic conditions, a membrane fraction was prepared from the cells. The membrane fraction was incubated with d7-sphingosine in the presence of C24:1-acyl CoA or C16:0-acyl CoA for 30 min at 37℃. Then, the quantities of d7-sphingosine-labeled C24:1-ceramide (C) and C16:0-ceramide (D) synthesized were determined by LC–MS. The relative levels of synthesis of d7-sphingosine-labeled ceramides were calculated as described in the Materials and Methods section. Data are presented as the means (n=3), and the dot plot indicates the values from each experiment. * p<0.05 (Student’s t-test).
Point 3: Please provide a discussion that could explain the biological importance of VL-SM under osmotic stress.
Response 3: To address this comment, we have added the following sentences to the Discussion section (Lines 486-499); “Although the biological importance of VL-SM under osmotic stress has not yet been elucidated, a previous study showed that an increase in very-long-acyl-chain sphingolipids in yeast cells may strengthen membrane integrity and endow some tolerance to hyperosmotic stress (Zhu et al, Appl. Environ. Microbiol., 2020). By analogy, the increased VL-SM synthesis in mammalian cells might be an acute response to adapt to hyperosmotic conditions. Considering that ceramides act as modulators of various proteins (Stith et al, JLR, 2019), the shift from L-ceramides to VL-ceramides might affect cellular signaling events. For example, C16-ceramide, but not other ceramide species, stabilizes the p53 tumor suppressor, a key regulator in various fundamental cellular events such as the cell cycle, apoptosis, and survival in response to diverse stimuli (Fekry et al, Nat. Comm., 2018). On the other hand, acute hyperosmotic stress induces caspase-mediated apoptosis (Thiemicke and Neuert, Sci. Adv., 2021). Thus, it is conceivable that a shift from L-ceramides to VL-ceramides may destabilize p53 and consequently attenuate p53-mediated hyperosmotic stress-induced cell death. Further studies will be needed to address these possibilities.”
Reviewer 2 Report
Shimasaki and collaborator presented work on CERT, a protein that delivers ceramide from the endoplasmic reticulum (ER) to the Golgi apparatus, where ceramide is converted to sphingomyelin (SM). CERT undergoes two independent phosphorylation events, the first concerns multiple phosphorylation in a serine-repeat motif (SRM), the second occurs at the level of the serine residue S315. The objective of the present work was to demonstrate that the two phosphorylation events are independent, and the authors exploited the observation obtained in a serendipitous manner that hyperosmotic stress evokes CERT S315 phosphorylation without affecting the SRM-phosphorylation state.
The authors clearly demonstrated that hyperosmotic stress induces S315 phosphorylation without affecting the SRM-phosphorylation state and that, under hyperosmotic conditions, the binding of CERT with VAP-A is enhanced in an S315 phosphorylation-dependent manner. Moreover, this binding increase occurs throughout the ER rather than restrictedly at the ER-Golgi MCSs.
I have just a few comments for the authors:
It is not clear why the authors chose the IP using the HA tag in monitoring the expression of HA-CERT, since CERT-KO cells do not express endogenous CERT. Hence, many of the WB results (Figures 1, 2, 3) could be conducted on the total lysate
To test the effect of hypoosmotic stress on S315-Cert phosphorylation, the authors incubated for an interval ranging from 10 min to 24h. This last time seems excessive to me. What is the vitality of the cells kept for such a long time?
What is the significance of hyperosmotic stress under physiological conditions?
It is not very clear which cell lines were used in the present work. The meaning and reason for using the HeLa-mCAT # 8 cell line have not been described. Likewise, the HeLA CERT KO / shCERT cell line and how it was obtained are not described in detail. Please indicate in which experiments the described cell lines were used and explain the rationale.
Author Response
Response to Reviewer 2 Comments
Point 1: It is not clear why the authors chose the IP using the HA tag in monitoring the expression of HA-CERT, since CERT-KO cells do not express endogenous CERT. Hence, many of the WB results (Figures 1, 2, 3) could be conducted on the total lysate.

Response 1: To detect the S315 phosphorylated form by western blotting analysis with anti-CERT S315p polyclonal antibodies, we had to concentrate CERT from the cell lysate fraction by immunoprecipitation (Figure 1D in the revised version). HA-CERT was immunoprecipitated with an anti-HA-epitope monoclonal antibody, but not with commercially available anti-CERT polyclonal antibodies. Thus, for this technical reason, we could not verify that hyperosmotic stress affects the level of phosphorylation of S315 of the endogenous CERT. We have added an explanation about this to the revised version of our manuscript (Lines 107-113).
Figure 1D in the revised version
Point 2: To test the effect of hypoosmotic stress on S315-Cert phosphorylation, the authors incubated for an interval ranging from 10 min to 24h. This last time seems excessive to me. What is the vitality of the cells kept for such a long time?
Response 2: The morphology of cells incubated under hypoosmotic conditions for 24 h appeared to be normal compared with the morphology of non-treated cells (Figure below, left). Moreover, the exposure of HeLa cells to hypoosmotic conditions for 24 h did not affect cell viability as determined by a WST assay (Figure below, right).
Point 3: What is the significance of hyperosmotic stress under physiological conditions?
Response 3: To address this comment, we have added the following sentences to the Discussion section (Lines 480-485).
“The osmolarity of the medium supplemented with 800 mM sorbitol was about 1,100 mOsm. The renal inner medulla is exposed to such severe hyperosmotic environments (Knepper, Kid. Int., 1982). SM molecular species with different acyl-chain lengths were shown to be distributed among different subregions of the kidney (Sugimoto et al, PLos One, 2016). It could be possible that the hyperosmotic stress-induced phosphorylation of CERT S315 partly underlies the region-dependent distribution of various SM molecular species in the kidney.”
Point 4: It is not very clear which cell lines were used in the present work. The meaning and reason for using the HeLa-mCAT # 8 cell line have not been described. Likewise, the HeLa CERT KO / shCERT cell line and how it was obtained are not described in detail. Please indicate in which experiments the described cell lines were used and explain the rationale.
Response 4: Thank you for your comment. To clarify which cell lines were used in each experiment, we have specified the names (CERT KO and CERT KO/shCERT) of the cell lines used in the respective figure legends.
We have also added the following sentence for explaining HeLa mCAT #8 cell in the Results section.
“To employ retroviral transfection, we used the HeLa mCAT#8 cell line, which stably expresses the mouse ecotropic retroviral receptor mCAT-1.” (Lines 80-81)
We have also added the following sentence to the Materials and Methods section, to briefly explain how the HeLa CERT KO/shCERT cell line was established, although further details will be published separately: “The HeLa CERT KO/shCERT cell line was established as described in a separate publication (Goto et al., submitted). In brief, the CERT KO cells were transfected using a retroviral vector containing a short hairpin RNA against CERT1 RNA to interfere with the expression of CERT1 mRNA. Then, a stable clone was isolated and used as the parental HeLa CERT KO/shCERT cell line.” (Lines 512-516)